# Growth of Nanocolumnar TiO_2_ Bilayer by Direct Current Reactive Magnetron Sputtering in Glancing-Angle Deposition Configuration for High-Quality Electron Transport Layer

**DOI:** 10.3390/mi14081483

**Published:** 2023-07-25

**Authors:** Perla Yanet Rosales Medina, Fernando Avelar Muñoz, Elida Flores Sigala, Roberto Gómez Rosales, Javier Alejandro Berumen Torres, José de Jesús Araiza Ibarra, Hugo Tototzintle Huitle, Víctor Hugo Méndez García, José Juan Ortega Sigala

**Affiliations:** 1Unidad Académica de Física, Universidad Autónoma de Zacatecas, Campus Universitario II, Av. Preparatoria S/N, Col. Hidráulica, Zacatecas 98068, Mexico; fernando.avelar@fisica.uaz.edu.mx (F.A.M.); elida.flores@fisica.uaz.edu.mx (E.F.S.); roberto.gomez@fisica.uaz.edu.mx (R.G.R.); javier.berumen@fisica.uaz.edu.mx (J.A.B.T.); araiza@fisica.uaz.edu.mx (J.d.J.A.I.); tototzintle@fisica.uaz.edu.mx (H.T.H.); jjosila@fisica.uaz.edu.mx (J.J.O.S.); 2CIACYT, Universidad Autónoma de San Luis Potosí, Av. Sierra Leona #550-2A, Col. Lomas de San Luis, San Luis Potosí 78210, Mexico; victor.mendez@uaslp.mx

**Keywords:** high-performance ETL TiO_2_, nanocolumns, glancing-angle deposition, GLAD, magnetron sputtering

## Abstract

The electron transport layer (ETL) plays a crucial role in solar cell technology, particularly in perovskite solar cells (PSCs), where nanostructured TiO_2_ films have been investigated as superior ETLs compared to compact TiO_2_. In this study, we explored the nanocolumnar growth of TiO_2_ in the anatase phase for bilayer thin films by DC reactive magnetron sputtering (MS) technique and glancing-angle deposition (GLAD). For the growth of the compact TiO_2_ layer, it was found that the crystalline quality of the films is strongly dependent on the sputtering power, and the samples deposited at 120 and 140 W are those with the best crystalline quality. However, for the nanocolumnar layer, the reactive atmosphere composition determined the best crystalline properties. By optimizing the growth parameters, the formation of TiO_2_ nanocolumns with a cross-sectional diameter ranging from 50 to 75 nm was achieved. The average thickness of the films exceeded 12.71 ± 0.5 µm. All nanostructured films were grown at a constant GLAD angle of 70°, and after deposition, the measured inclination angle of the nanocolumns is very close to this, having values between 68 and 80°. Furthermore, a correlation was observed between the quality of the initial layer and the enhanced growth of the TiO_2_ nanocolumns. All bilayer films are highly transparent, allowing light to pass through up to 90%, and present a band gap with values between 3.7 and 3.8 eV. This article offers the experimental parameters for the fabrication of a nanocolumnar TiO_2_ using the magnetron sputtering technique and the glancing-angle deposition configuration.

## 1. Introduction

The importance of research on perovskite solar cells is strongly influenced by the notable improvement in their energy conversion efficiency (PCE), given that reports indicate that in just over 10 years, they have had an increase in their PCE of 3.8% to 25.7% [1,2,3,4]; however, the Shockley–Queisser limit for single-junction perovskite solar cells is ≈33%, it also presents an extension to multijunction, depends on the junction it could be from 44% to 65% [5,6,7]. The typical structure of a perovskite solar cell consists of an electron transport layer (ETL), an active material perovskite layer, and a hole transport layer (HTL). Each layer is very important, but, in practice, the loss of efficiency is closely related to the ability of the ETL to transport photogenerated electrons. The electron transport layer plays an important role within the structure of the solar cell, since the electron transport layer, depending on its morphology, the type of material from which it is built, and its physical properties, will be used for better stability, prolonged life, and offer higher PCE, some of the most widely used materials are metal oxides, some of the most commonly used materials are metal oxides, such as TiO_2_ [8,9,10], ZnO [11,12,13], SnO_2_ [14,15,16], SiO_2_ [17], and ZrO_2_ [18,19]; each material provides specific advantages. 

In this context, recent studies indicate that nanostructured ETLs TiO_2_ offer greater stability, longer useful life, and a higher percentage of energy conversion. An electron transport layer must comply with some requirements: (1) well-matched energy alignment to trigger electron transfer while blocking holes; (2) high transparency to allow efficient light harvesting; and (3) excellent electron mobility to minimize charge accumulation [20,21,22,23,24]. TiO_2_ offers high performance as an ETL in PSCs due to its alignment of the TiO_2_ conduction band with the perovskite layer [25,26,27]; furthermore, the percentage of transparency of TiO_2_ is greater than 85% in practically all spectral windows [28].

The TiO_2_ is an n-type semiconductor [29], is chemically very stable, absorbs electromagnetic radiation near the UV region, and its band gap is around 3 eV [30]. Titanium oxide has four crystalline phases: anatase, rutile, brookite, and a high-pressure type α Pb O_2_. The rutile phase is the only one that is thermodynamically stable [31], while the other two, anatase and brookite, are metastable and transform into rutile when heated [32]. However, for the use of TiO_2_ as an ETL layer, it has been shown that a mixture of rutile and anatase TiO_2_ crystals could improve solar conversion efficiency [33,34,35,36].

The growth parameters of thin films using the magnetron sputtering technique depend on several factors, including the target size, the distance between the target and the substrate surface, and the pressure during the growth process. Typically, adjustments are made to the deposition power and growth atmosphere to achieve desired results. The power of the deposit is not fixed; in fact, there is a great dependency between this value and the characteristics of each sputtering system. There are articles reporting a power deposition of 100 to 1000 W [37,38,39,40] and a very variated reactive atmosphere of oxygen/argon with proportions of 2/8 to 5/35 sccm [40,41,42,43,44]. 

According to the literature, the geometry of ETL plays an important role in PSCs to maximize device performance. The implementation of TiO_2_ nanocolumns as an ETL presents great advantages compared to flat and mesoporous layers when constructing a PSC [27,33,34,35], as it provides greater stability and efficiency by means of some advantages: (i) an enhanced contact zone between perovskite and ETL, and increased exciton generation; (ii) the 1D morphology offers a directive path for electron transport to the TCO substrate and minimizes resistance to charge transport; and (iii) larger nanorods improve the light scattering and absorption properties of PSCs [45]. It has been found that the presence of semiconductor metal oxides is not unique.

There exist a variety of methods to deposit compact and nanostructured TiO_2_ from chemical (CVD) and physical (PVD) techniques; however, the deposition of thin films of transparent oxides by sputtering offers excellent quality properties in these materials, with a low cost. There is a huge research gap around the preparation of nanomaterials through the sputtering technique; in fact, there are very few reports of thin film growth using GLAD in sputtering. This work offers an accurate report of nanocolumnar growth of TiO_2_ using DC magnetron reactive sputtering in GLAD configuration, where such nanocolumns offer the essential morphological and superficial characteristics that a high-quality ETL requires.

## 2. Materials and Methods

### 2.1. Compact TiO_2_ Test Layer Deposition

As an initial procedure, three compact test layers were deposited using the reactive DC magnetron sputtering technique in a normal (parallel) configuration. A pure titanium sputtering target (99.999%, from Kurt J. Lesker) of 2″ diameter was used as the precursor. Before each deposit, the sputtering chamber was evacuated to a base pressure of 2.0 × 10^−6^ Torr, and the depositions were performed at a pressure of 3.0 × 10^−3^ Torr using a reactive sputtering atmosphere composed of a mixture of 10 sccm of oxygen (99.999%) and 40 sccm of argon (99.995%) (10 O_2_/40 Ar sccm), as described in the literature for nanocrystalline deposition TiO_2_ [40]. For these three test films, the parameter that was varied for compact test layers deposited was the deposition power, with values of 100, 120, and 140 W applied to Si substrates (100). 

Given the potential for slight variations in the characteristics of the deposited films in different sputtering systems, these compact films were specifically deposited to identify the deposition parameters that would yield the highest quality in terms of crystallinity. Additionally, these films served as an essential experimental reference for the subsequent formation of our nanostructured bilayer films, with the first layer designed to exhibit a compact crystalline structure. 

### 2.2. Nanocolumnar TiO_2_ Bilayer Growth

The growth of nanostructured TiO_2_ bilayer films was developed using the sputtering technique with a reactive DC magnetron, and a pure titanium sputtering target (99.999%, from Kurt J. Lesker) of 2″ in diameter was used as precursors. Before each deposit, the sputtering chamber was evacuated to a base pressure of 2.0 × 10^−6^ Torr, and the depositions were performed at a pressure of 3.0 × 10^−3^ Torr using a reactive sputtering atmosphere composed of a mixture of argon (99.995%) and oxygen (99.999%) with different proportions of them for each prepared sample. The growth of each sample was carried out in three stages according to the following description: First, a thin film of TiO_2_ was deposited to obtain the compact layer of TiO_2_ according to the conditions previously established in the test films. Second, after the deposit of the first layer, the films were annealed at 400 or 500 °C to improve their crystalline quality and heat treatment was administered on a simple plate exposed to the environment and at atmospheric pressure. Finally, once the annealing treatment was carried out on the monolayer film, the columnar layer of TiO_2_ was grown on it, using the magnetron reactive sputtering technique under the GLAD configuration. All the deposits were performed simultaneously using two different substrates, namely glass plain slide Pearl brand and silicon (100), and in order to remove any contamination from the target, a pre-sputtering of between 5 and 20 min was performed, depending on the time it took for the plasma to reach stability. And for all samples, the deposition temperature was maintained at room temperature (around 22 °C). 

From the growths carried out, the three most outstanding results were selected, that is, the M3, M4, and M6 samples, whose growth parameters were the following: 

For sample M3, a planar film was deposited using an atmosphere of 15 O_2_/40 Ar and a sputtering power of 140 W. This monolayer film received a thermal treatment of 500 °C for one hour, then on this compact film, a nanocolumnar TiO_2_ thin film was grown, using the magnetron sputtering technique implementing the GLAD configuration at an angle of inclination of 70°, for 90 min, with a sputtering power of 120 W, maintaining the vacuum pressure constant at ~10^−3^ Torr and under a sputtering atmosphere of 10 O_2_/40 Ar sccm.

For sample M4, a compact film with an atmosphere of 5 O_2_/40 Ar and a deposit power of 140 W was deposited. This monolayer film received an annealing treatment of 500 °C for one hour, then, on this compact film, a thin nanocolumnar film was grown, made by TiO_2_ using the magnetron sputtering technique implementing the GLAD configuration at an angle of inclination of 70°, for 90 min, under a sputtering atmosphere of 5 O_2_/40 Ar and a deposit power of 140 W. The last sample, named M6, this sample consists of a planar film with an atmosphere of 15 O_2_/40 Ar and a deposit power of 120 W. This monolayer film received a thermal treatment of 400 °C for one hour, then, on this compact film, a nanocolumnar thin film was grown, made by TiO_2_ using the magnetron sputtering technique implementing the GLAD configuration at an angle of inclination of 70°, for 90 min, with a sputtering power of 120 W, under an atmosphere of 10 O_2_/40 Ar sccm and a sputtering power of 120 W (Table 1 presents the deposition conditions mentioned above). 

### 2.3. Characterization

For the characterization of the samples obtained, the samples prepared on a glass substrate were then used for UV–Vis spectroscopy, and the samples prepared on silicon were used for XRD, FTIR, and SEM studies. X-ray diffraction characterization was performed using Rigaku Dmax2100 equipment by Rigaku Corporation, Tokyo, Japan. The diffractometer is equipped with a vertical goniometer (185 mm), a Ni filter, and a scintillation counter. The radiation generator tube is made of copper (K*α* = 1.5406 Å), and the maximum power is 2 kW (50 kV and 40 mA); however, the measurements in the materials use 30 kV and 20 mA (600 W). The irradiation zone in the sample was controlled by the divergence grating. The step size resolution of the goniometer is 0.005°. A second analysis was performed with a PerkinElmer Spectrum GX Fourier-transform infrared (FTIR) spectrophotometer coupled with an AutoIMAGE microscope. The FTIR operates in the mid- and near-infrared (MIR and NIR) and has four accessories. The one used to measure the samples was the total attenuated reflectance (ATR) with a diamond tip and zinc selenide (ZnSe). The ATR with ZnSe has a controlled heating system from room temperature to 120 °C. Another optical characterization from which we measure the transmission of light corresponds to the transmission percentage (%T) that was obtained from a PerkinElmer UV–Vis ellipsometer, precisely model Lambda 35, which operates at wavelengths of 200 to 1100 nm. The last characterization was carried out to know the morphological characteristics; the morphology was analyzed by field emission scanning electron microscopy (FE-SEM-TESCAN MIRA 3 model by TESCAN ORSAY HOLDING, Brno, Czech Republic).

## 3. Results

Figure 1 shows the cross-section of the complete structure of one of the grown bilayer samples. In this image, it is possible to observe three regions, the lower part corresponds to the substrate, which in this case is Si (100), and the intermediate region corresponds to the compact crystalline TiO_2_ layer that was deposited under normal sputtering conditions. Finally, the upper region corresponds to the columnar TiO_2_ layer, which was grown under sputtering conditions of the glazing angle. In general, the compact layer has a thickness of ~2 μm, while the columnar layer has an average thickness of several micrometers and presents a strong dependence on the deposition conditions, such as the composition of the sputtering atmosphere and the sputtering power used, in addition to the fact that the surface on which the bilayer samples were structured is different depending on the deposition conditions of the first layer.

### 3.1. Compact TiO_2_ Test Layers

The diffractograms in Figure 2 show typical TiO_2_ X-ray diffraction patterns corresponding to the anatase phase. (The diffraction planes of the samples were identified from the XRD patterns, and correspondence was found with the #21-1272 file reported in the JADE^®^ program from MDI, Livermore, CA, USA). In addition to the anatase phase, were identified the rutile peak (211), which is located at 2*θ* = 54.1° in each of the samples, this related to a combination of crystalline phases. 

For the three compact TiO_2_ test samples that were deposited under the same sputtering atmosphere conditions (10 O_2_/40 Ar) under normal sputtering conditions, varying only the deposit power to 100, 120, and 140 W. The diffraction patterns showed a diffraction peak at an angle 2*θ* = 24.8° as the principal maximum, and this peak corresponds to the signal of the plane (101) of the structure tetragonal anatase typical of TiO_2_. The film deposited at 140 W shows a better definition of the crystallographic planes, and in it, in addition to this maximum, it is possible to identify the peaks associated with the (004), (200), (105), and (211) planes at 2*θ* equal to 37.5°, 47.7°, 53.7°, and 55.1°, respectively. The effect of the deposition power in the structure can be observed in the XRD pattern obtained for the compact films. In Figure 2, with increasing deposition power, the crystalline quality improves, which is reflected by the sharper and higher diffraction peaks obtained. The justification for this result is that the intensity of the peaks in a diffraction pattern is directly associated with the size of the crystallite, and a taller and sharper diffraction peak corresponds to a larger size of the crystallite, while when the size of the crystallite is of nanometer size, the peaks are expected to be of low intensity and with a higher width of medium height. For this reason, conform increases the sputtering power in the deposition, and the size of the crystallite in thin films increases; for this reason, the diffraction peaks are observed with better definition and greater intensity. On the other hand, the shift of the diffraction peaks is an indication that the crystalline structure is stressed, due in this case to defects in its formation; however, as can be seen in the diffractograms, the highest intensity peak, which corresponds to the plane (101) of the anatase structure of TiO_2_, remains centered in the same position for all three samples.

The results of ATR spectroscopy are shown in Figure 3. In this characterization, two absorption bands can be identified that correspond to the Ti–O–Ti bond when there is an incidence of light with a wave number of 734 cm^−1^ [43,46] and a bond of Si–O for a wavenumber of 1106 cm^−1^. An additional signal at 1240 cm^−1^ can be observed in the ATR spectra for the film deposited at 100 and 140 W; typically, it is associated with the C–O bond. 

### 3.2. Nanocolumnar TiO_2_ Bilayers

The diffractograms obtained from the XRD characterization for the three bilayer samples are shown in Figure 4. The M3 and M6 films present a phase combination where the dominant crystalline structure was identified as rutile, a phase identified by the crystallographic planes (110), (101), and (220); these crystallographic planes correspond to 2*θ* = 27.9°, 42.5°, and 56.7°, respectively. In addition, these samples exhibit an anatase phase identified by crystallographic planes (101), (004), (112), (200), and (105) at 2*θ* = 25.0°, 37.7°, 38.4°, 47.9°, and 53.8°, respectively. In the case of M4, the anatase phase is dominant; the corresponding diffractograms present the same crystallographic planes as M3 and M6 but show a higher intensity and better definition of the diffraction pattern corresponding to the anatase phase.

The three samples are observed to have a combination of anatase and rutile phases, but the nanostructured TiO_2_ bilayer denoted M4 presents the proportions of the mixture of anatase and rutile phases, which meets the optimization requirements in terms of the crystalline phases reported in the literature. 

XRD measurements allowed one to determine the crystallite size (*D*) and lattice parameters. The value of (*D*) was calculated from the Scherrer equation:(1)$$D=\frac{k\lambda }{\beta \cos\theta_{\beta }}$$

The interplanar distance (*d_hkl_*) of the main peaks was determined by Bragg’s law:
(2)$$d_{hkl}=\frac{\lambda }{2\sin\theta_{hkl}}$$
where *d_hkl_* is the interplanar distance; *h*, *k*, and *l* are the Miller indices; and the lattice constants *a* and *c* were estimated using the equation, which corresponds to the separation of planes of a tetragonal structure.
(3)$$\frac{1}{d_{hkl}^{2}}=\frac{h^{2}+k^{2}}{a^{2}}+\frac{l^{2}}{c^{2}}$$

To calculate the crystalline parameter *a* of the anatase phase and the interplanar distance *d*_200_, the signal identified by the crystallographic plane (200) was used, while the parameter *c* and the interplanar distance *d*_101_ were calculated from the peak (101). As can be seen in Figure 4, the both peaks are identified as being the two of the highest intensity and the most important to identify the anatase phase of TiO_2_. The obtained crystallographic parameters corresponding to the anatase phase are showed in Table 2. 

Similarly, to calculate the crystalline parameter *a* of the rutile phase and the interplanar distance *d*_110_, the signal identified by the crystallographic plane (110) was used, while the parameter *c* and the interplanar distance *d*_101_ were calculated from the peak (101). The obtained crystallographic parameters corresponding to the rutile phase are showed in Table 3.

The vibrational properties of the deposited TiO_2_ films were analyzed by Fourier-transform infrared (FTIR) spectroscopy. The results obtained are shown in Figure 5. In these spectra, the image shows the appearance of an absorption band at a wave number of 638 cm^−1^, which corresponds to the Ti–O–Ti bond of TiO_2_ [46,47]. 

Figure 6 shows the UV–Vis spectrum of TiO_2_ bilayer films. From the transmission spectra, it can be seen that most of the samples present a maximum light transmission in the range of *λ* of 350–600 nm, and the maximum transmission percentage reaches values between 90% and 95%. Figure 7 was created using the Tauc method, which allows the determination of the optical band gap. The growth of the Tauc curve becomes linear when the photon energy aligns with the bandgap value. 

The optical band gap, denoted as *E_g_*, was determined using Tauc’s method [48] and can be calculated using the following equation:
*αhν* = *A*(*hν* − *E_g_*)*^n^*
(4)

where *A* is a constant; *hν* is the photon energy; and the exponent is *n* = 1/2 for allowed direct, *n* = 2 for allowed indirect, *n* = 3/2 for forbidden direct, and *n* = 3 for forbidden indirect transitions [49]. It is well known that TiO_2_ films have allowed direct transitions (*n* = 1/2). The band gap is determined by extrapolating the linear portion of the curves to zero absorption. The linear behavior of the graph indicates that (*αhν*)^2^ has a linear dependence on the energy, for this reason, obeying the proposed Tauc equation, and for when *α* = 0, then *hν* = *E_g_*. 

As can be seen in both figures, the three deposited bilayers have very high transparency (greater than the 85% required) in the entire visible region, with a window of transparency that enters deeper in the UV and in the NIR than the typical behavior for other materials used for the manufacture of ETLs such as ZnO or SnO. 

To observe the nanocolumn structure of the TiO_2_ films, a morphological characterization was carried out using the scanning electron microscopy technique, both on the surface and on the profile of some of the deposited samples. 

In Figure 8, two different micrographs of the same sample with the best morphology of all our samples. In the lower micrograph, an SEM image of the profile (cross-section) is observed where the angle of inclination for nanocolumnar growth is identified. The upper image corresponds to the superficial image but with contrast enhancement to better identify the aforementioned areas; additionally, a graduated ruler is added to the image to facilitate the approximation of the thicknesses and the angle of inclination of the nanocolumns. The lower micrograph shows an SEM image of the sample surface in which a distribution of uniform islands formed from the nanocolumnar growth of the second layer is observed. Figure 9 shows the SEM profile of the TiO_2_ nanocolumnar layer. In the three samples, it is possible to observe defined filaments; the fact of having filaments ensures that the layer is not compact and, in addition, this, together with the superficial SEM images, allows us to ensure a nanocolumnar growth since we are seeing the condition satisfied to have a growth on the nanoscale, that at least one of its dimensions is below 100 nm. It can be observed that there is a direct correspondence relationship between the deposition angle of the nanocolumns and the inclination they exhibit. When a deposition angle of 70 degrees is considered, nanocolumns inclined at values close to this angle are observed. The measured angles on the inclination of the nanocolumns range from 68 to 80 degrees, and in many other studies, the correlation between the angle of deposition in GLAD and the angle of columnar growth is reported [49,50]. This image is of great value, as it provides evidence of the existence of the columns, which are closely related to the crystalline quality of the substrate. 

Figure 10 corresponds to the surface SEM images of the bilayer TiO_2_ films. The three micrographics show small circles with an approximate diameter of 70 nanometers. This image, together with the cross-sectional SEM image 8, allows us to confirm the existence of nanocolumns grown using the magnetron sputtering technique with the oblique angle deposition technique.

Histograms to compare the diameters of the nanocolumns for each of the samples are shown in Figure 11. It can be observed that the average diameter of each of the bilayer films shows a significant variation. This variation will be explained in this study’s discussion, where factors such as film composition, deposition parameters, and annealing treatment, among others, will be considered. These factors can influence the formation and growth of nanocolumns, which in turn affects their average size. 

## 4. Discussion

In this research, the effort has focused on improving the properties of TiO_2_ ETLs, including morphological control and surface modification. Of these two mechanisms, the morphology control of the ETL has demonstrated that this is an easier form of improving electron extraction and transport by inhibiting carrier recombination and promoting charge extraction. The nanocolumnar morphology of the ETL surface can lead the perovskite material to enter between the nanocolumns and cause a larger surface area of the perovskite layer. 

According to the crystalline properties of the bilayer films, it was found that the sample with the best crystalline quality is sample M4. The high crystallinity in the nanostructured layer was associated with the superior crystalline quality of the compact layer. Within the scientific literature, it is becoming increasingly clear that the epitaxial growth in multilayer films [51] is favored by the crystalline quality of the first layer. Evidence of this superior crystalline quality is seen both in the XRD patterns and in the IR analysis, where the absorption band corresponding to the Ti–O–Ti bond is better defined for this sample, and the XRD peak associated with the (101) direction presents a higher intensity compared to the other two samples. In addition, in the growth of the nanostructured layer, the oblique angle technique was chosen to grow the nanocolumns under the same parameters as those of the compact layer, the highest intensity in the XRD pattern of the compact layer was observed for the film deposited at a higher sputtering power. 

For samples M3 and M6, their nanocolumnar layer was deposited under identical parameters. However, there were a couple of differences in the compact layer that influenced the results. Both were grown under the same sputtering atmosphere conditions but with a variation in the deposition power (140 and 120 W, respectively). Furthermore, they underwent thermal treatment at 500 °C and 400 °C, respectively. In this case, we believe that the 400 °C thermal treatment improved the crystalline quality of sample M6, resulting in a nanocolumnar layer with better crystallinity compared to M3. Regarding the intensity of the band associated with TiO_2_, M3 exhibits a higher intensity. The morphology of the films was examined by surface and cross-sectional SEM analyses. In the cross-sectional SEM image, it is evident that sample M4 exhibits a more defined filament structure. Overall, this work confirms the findings of previous researchers on the relationship between the deposition angle and the inclination of the nanocolumns. Although the deposition angle was not altered in this study, this effect has already been discussed in other articles [49,50]. The superficial SEM images for samples M3 and M6 exhibit a better distribution of the nanocolumns, in terms of the separation between the nanocolumns being less compared to M6. For M3, the nanocolumns have an average diameter of 43.9 nm, while for M6, the average diameter is 73.8 nm. 

According to the superficial morphology of the bilayer films, in the SEM images of samples M3 and M6, homogeneity is observed in the distribution of the diameters of the nanocolumns. As for sample M4, despite having better physical properties, it exhibits inhomogeneity in the distribution and diameters of the nanocolumns. However, the average diameter of the nanocolumns is 66.8 nm. This behavior can be attributed to the oxygen concentration during the deposition of the nanocolumnar layer, but at this moment there is not enough evidence; however, there will be an opportunity to continue this investigation. All three layers are highly transparent, allowing light to pass through up to 90%. However, in this case, samples M4 and M6 are more transparent in the range of 350 to 600 nm, while M3 is more transparent in the range of 600 to 1000 nm, and present a band gap with values between 3.7 and 3.8 eV; therefore, it fully complies with the optical condition to be used as ETL. 

After analyzing the physical properties of the films, such as crystallinity, molecular vibration, transparency under visible light, and morphological characteristics, it is appropriate to conclude that the three samples meet the requirements to be considered a high-quality electron transport layer (ETL). In conventional perovskite solar cells, TiO_2_ is widely used as an ETL due to its favorable energy alignment [52,53,54]. In this case, of the three samples, sample M4 exhibits the ideal mixture of crystalline anatase and rutile phases of TiO_2_. Furthermore, the observed nanocolumnar growth makes it a promising candidate for the implementation of ETL, as this structure improves electron mobility and suppresses surface recombination [28]. 

## 5. Conclusions

This study concludes that nanocolumnar growth on bilayer films of TiO_2_ can be achieved through the direct current reactive magnetron sputtering technique in the glancing-angle deposition (GLAD) configuration. The columnar TiO_2_ growth, with optimal characteristics, can be obtained using sputtering powers between 120 and 140 W, low oxygen concentrations, and appropriate thermal treatment of the compact layer. Additionally, this study reveals that the best results for columnar growth by MS in the GLAD configuration are favored by the crystalline quality of the compact layer; however, the crystalline phase of the compact layer is not sufficiently decisive to maintain the growth of the nanocolumns in the same phase. Moreover, all the grown nanostructured TiO_2_ bilayers are highly transparent, allowing light to pass through up to 90% in the range of 350 to 600 nm and present a band gap with values between 3.7 and 3.8 eV. Finally, after analyzing the physical properties of the films, such as crystallinity, molecular vibration, transparency under visible light, and morphological characteristics, it is appropriate to conclude that of the three samples, sample M4 meets all the optical, morphological, and superficial requirements to be considered for applications as a high-quality electron transport layer.

## Figures and Tables

**Figure 1 micromachines-14-01483-f001:**
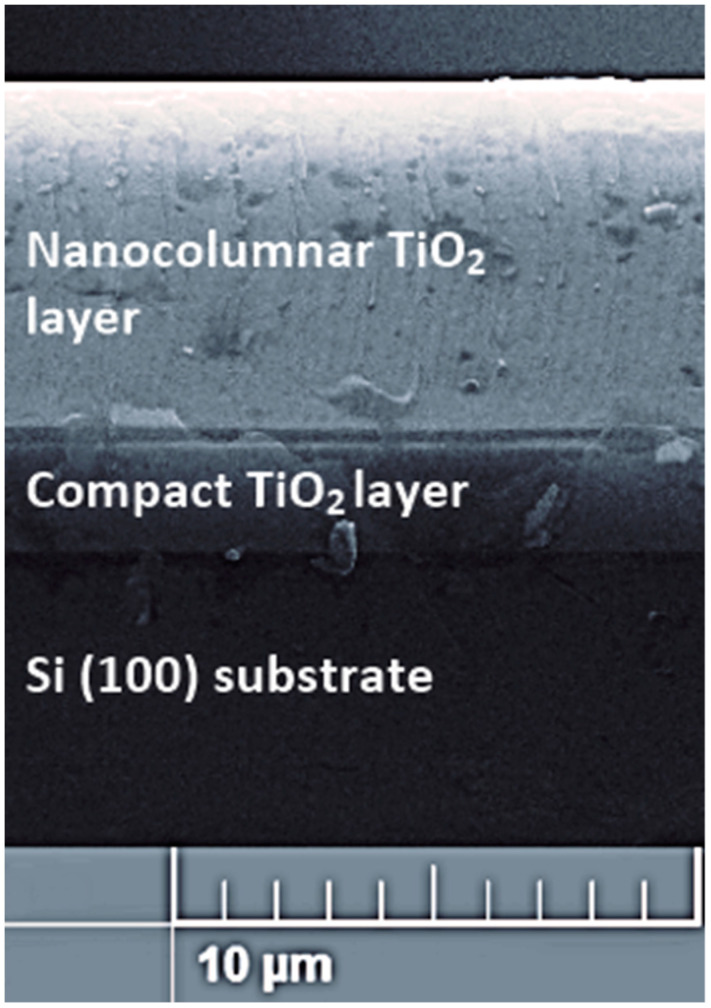
Cross-sectional SEM image of the complete structure of TiO_2_ bilayer.

**Figure 2 micromachines-14-01483-f002:**
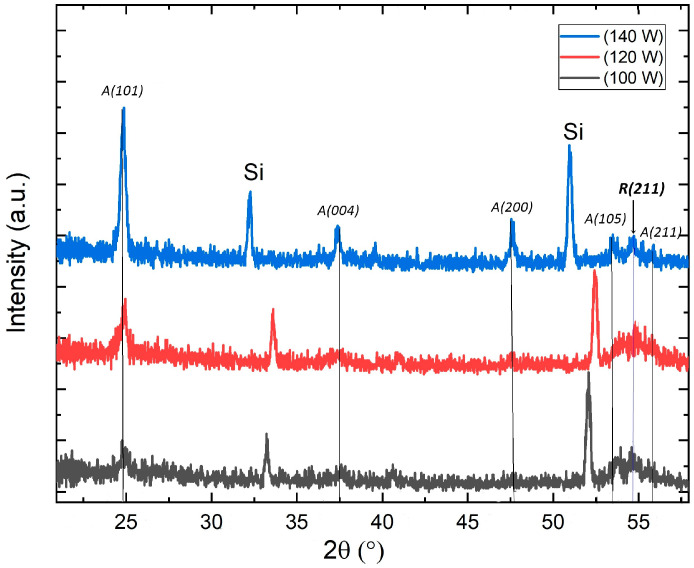
XRD patterns corresponding to test samples (described in Section 2.1). The X-ray diffraction patterns of the three compact TiO_2_ test layers deposited over Si (100) substrates, under the same sputtering atmosphere composition 10 O_2_/40 Ar sccm and for different sputtering powers.

**Figure 3 micromachines-14-01483-f003:**
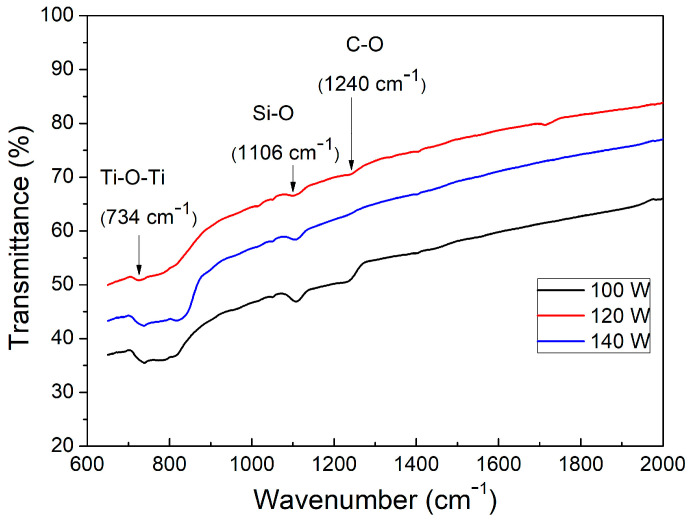
ATR pattern corresponding to test samples (describe in Section 2.1). ATR spectroscopy of the three compact TiO_2_ test layers deposited over Si (100) substrates, under the same sputtering atmosphere composition 10 O_2_/40 Ar sccm and for different sputtering powers.

**Figure 4 micromachines-14-01483-f004:**
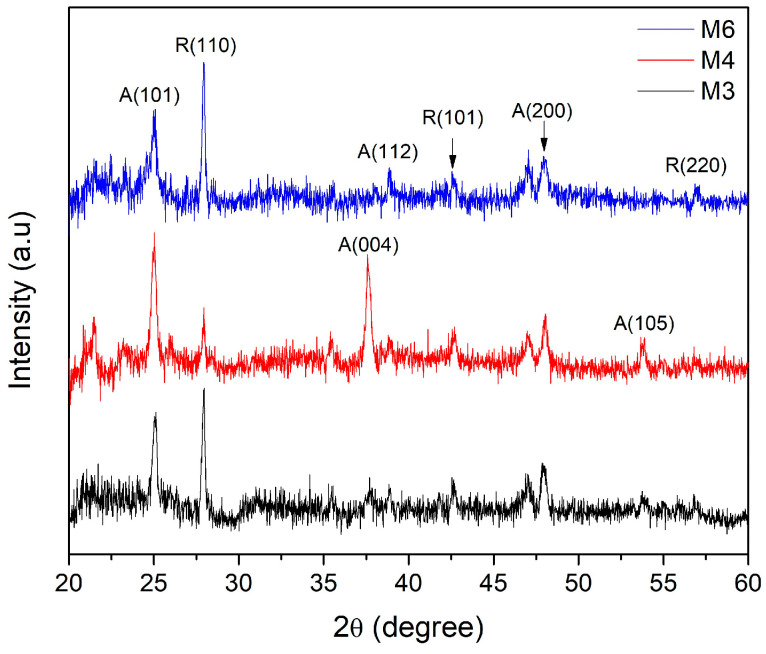
X-ray diffraction patterns of the complete nanostructured TiO_2_ bilayer deposited on Si (100). In the three samples, a combination of anatase and rutile phases can be observed. M3 and M6 exhibit dominance in the rutile phase; on the other hand, M4 presents dominance in the anatase phase.

**Figure 5 micromachines-14-01483-f005:**
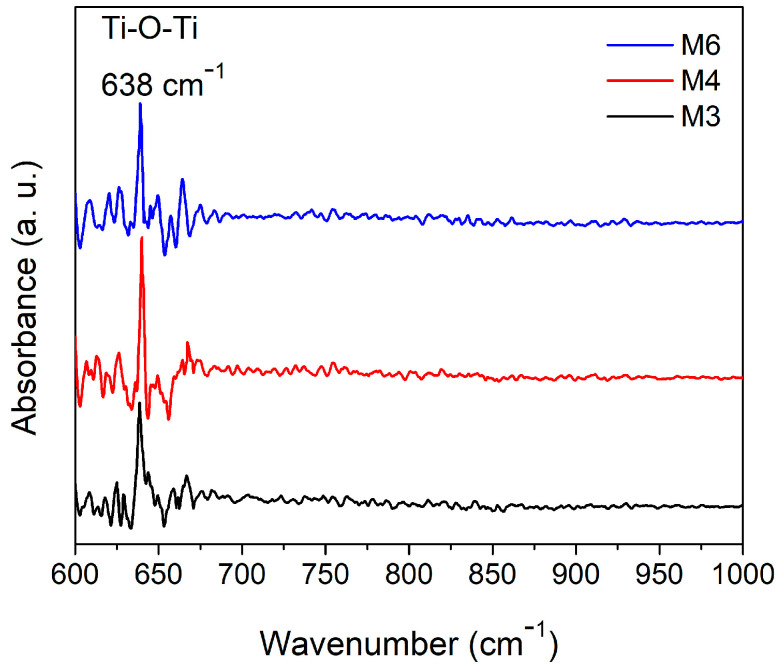
FTIR spectra of the complete nanostructured TiO_2_ bilayer deposited on Si (100).

**Figure 6 micromachines-14-01483-f006:**
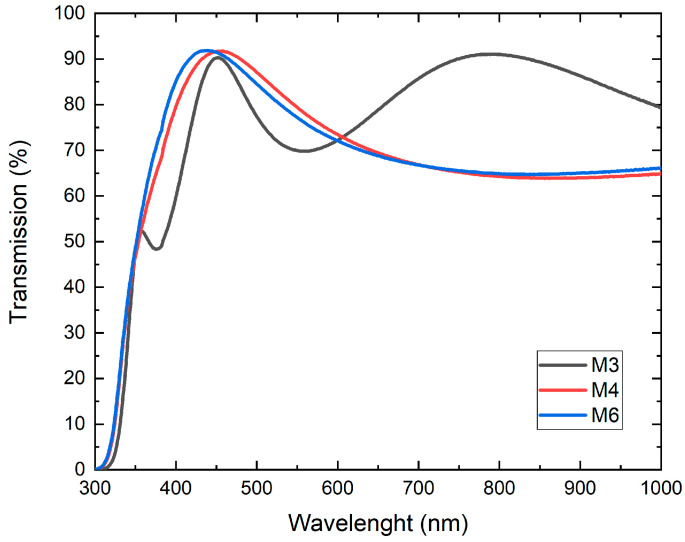
UV–Vis spectroscopy for nanostructured TiO_2_ bilayer films. Figure shows a relationship between wavelength and percentage transmission.

**Figure 7 micromachines-14-01483-f007:**
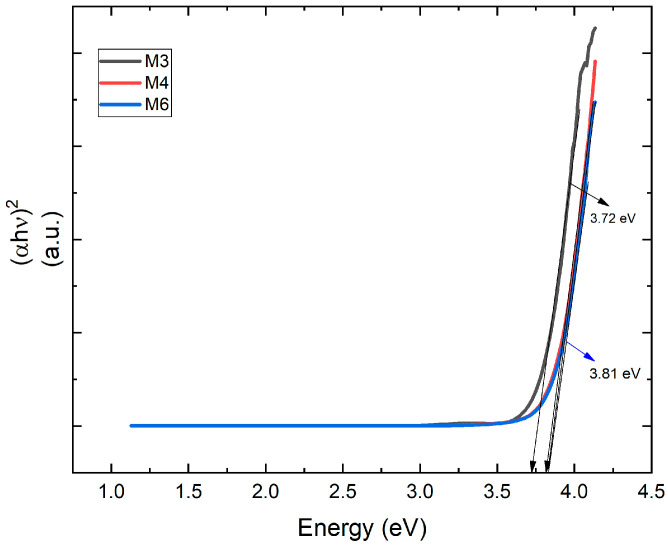
Tauc approximation to determine the value of the band gap of the bilayer films of TiO_2_.

**Figure 8 micromachines-14-01483-f008:**
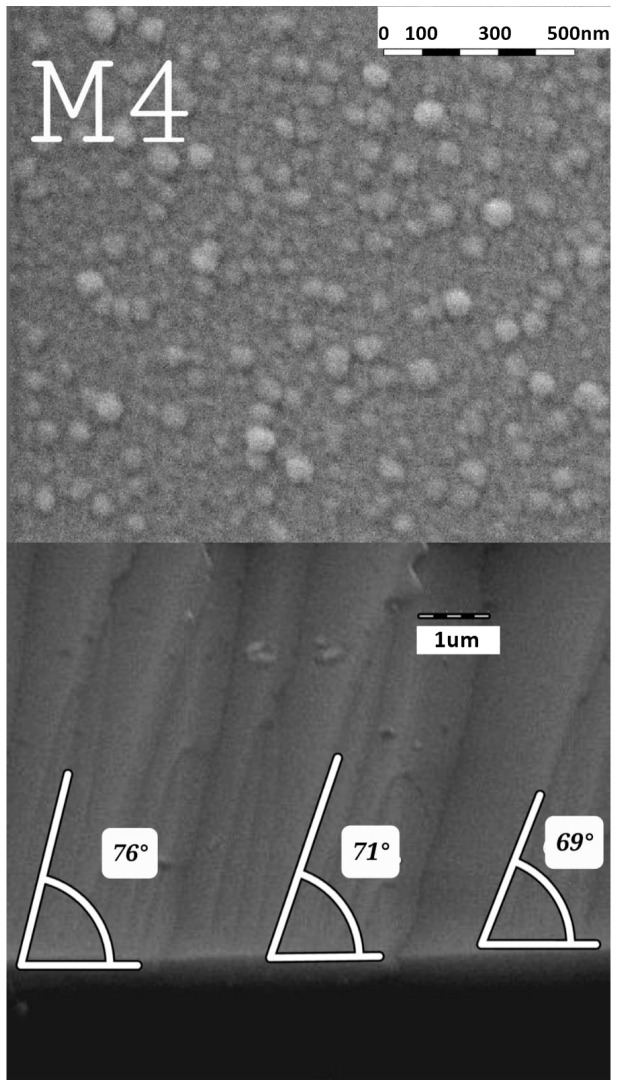
Superficial and cross-sectional micrographs of the same sample. This SEM analysis shows us cross-sectional and superficial morphology of the sample M4.

**Figure 9 micromachines-14-01483-f009:**
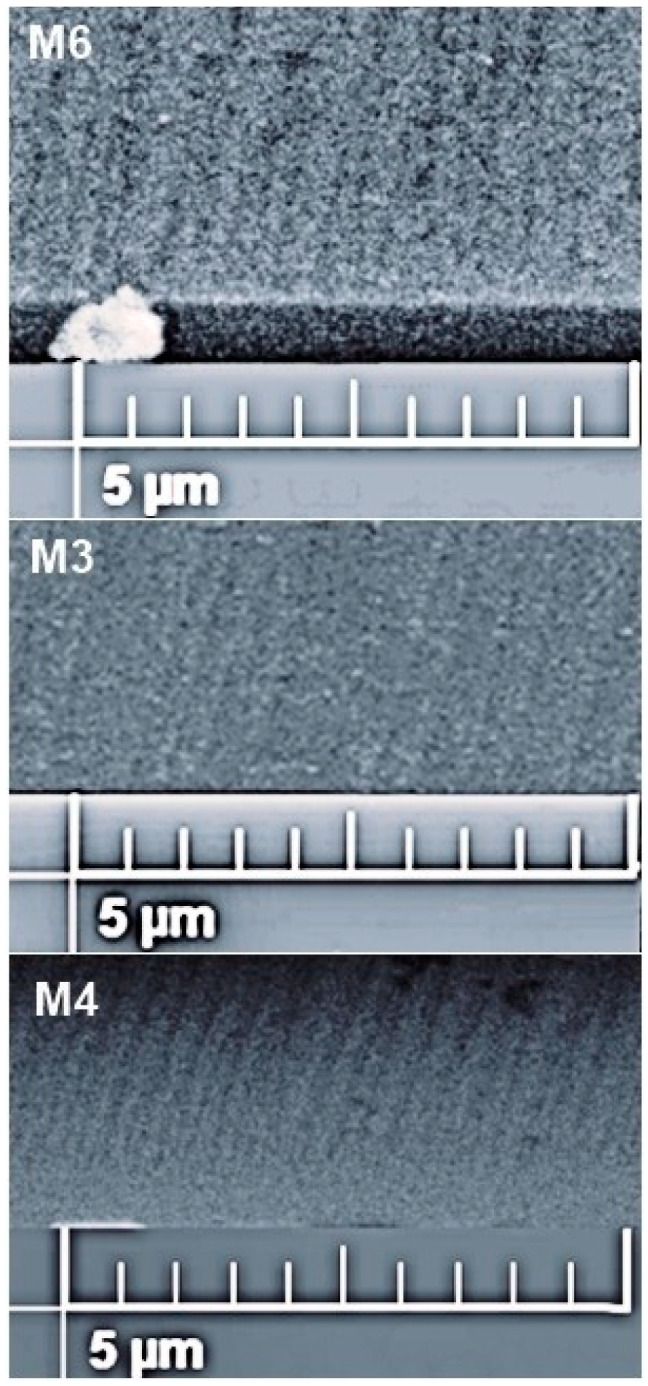
Cross-sectional SEM images of the anatase nanocolumns TiO_2_ in the bilayer films.

**Figure 10 micromachines-14-01483-f010:**
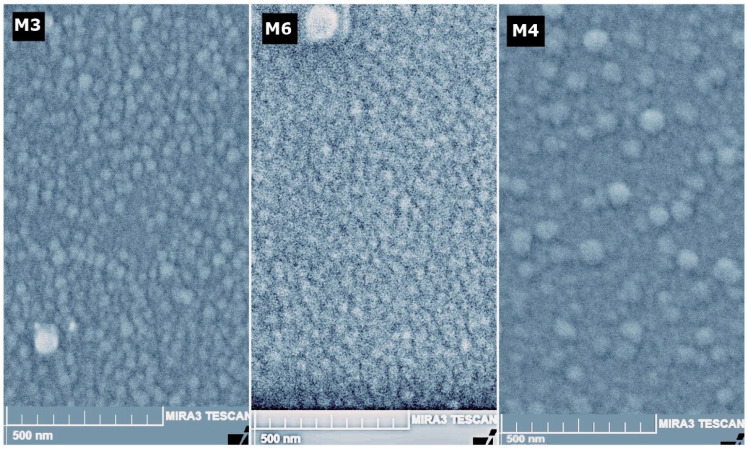
Surface morphology SEM images of the three nanocolumnar TiO_2_ bilayers. As shown in the images, on the surface of samples M3 (**left**) and M6 (**center**), the surfaces of the columns are observed with a uniform distribution in size and shape, for the surface of sample M4 (**right**), a formation of larger islands can be seen.

**Figure 11 micromachines-14-01483-f011:**
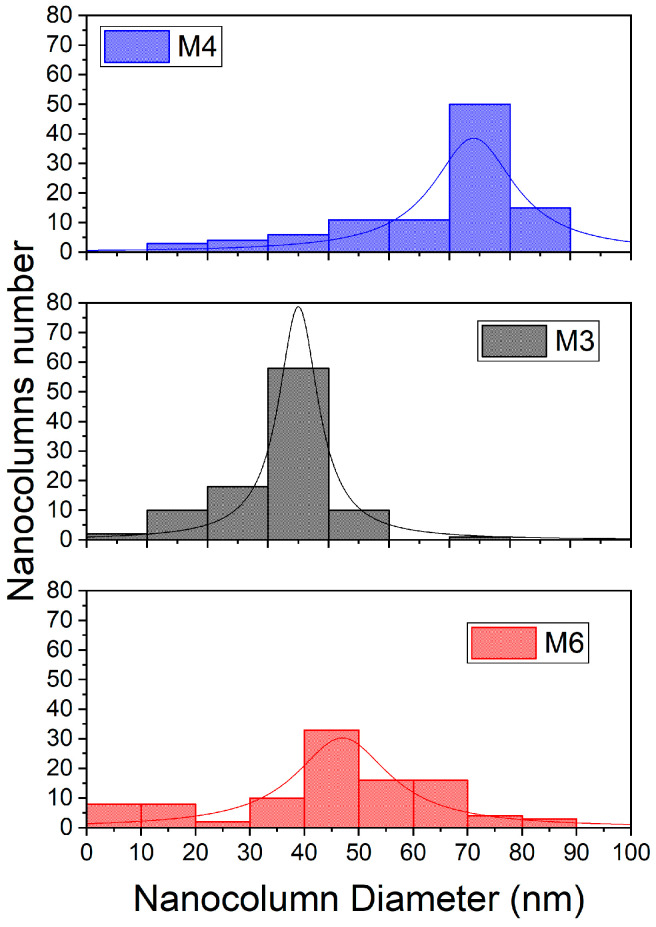
Histogram comparing the average diameter measured for 100 nanocolumns in each of the samples.

**Table 1 micromachines-14-01483-t001:** Experimental parameters used for the formation of the nanostructured TiO_2_ bilayers deposited by normal MS and MS on GLAD configuration.

Sample	Compact TiO_2_ Layer			Nanocolumnar TiO_2_ Layer	
	Atmosphere O_2_/Ar (sccm)	Power (W)	Thermal Treatment (°C)	Atmosphere O_2_/Ar (sccm)	Power (W)
M3	15/40	140	500	10/40	120
M4	5/40	140	500	5/40	140
M6	15/40	120	400	10/40	120

**Table 2 micromachines-14-01483-t002:** Crystallographic data corresponding to the anatase phase of TiO_2_ films with nanocolumnar growth.

Sample	2*θ* (°)	*β* (°)	*D* (Å)	*d*_101_ (Å)	*a* (Å)	*c* (Å)
M3	25.05	0.2847	285.8	3.55	3.79	10.16
M4	24.99	0.3725	218.4	3.55	3.78	10.22
M6	24.98	0.5176	157.2	3.56	3.78	10.23

**Table 3 micromachines-14-01483-t003:** Crystallographic data corresponding to the rutile phase of TiO_2_ the films with nanocolumnar growth.

Sample	2*θ* (°)	*β* (°)	*D* (Å)	*d*_110_ (Å)	*a* (Å)	*c* (Å)
M3	27.92	0.2124	385.40	3.19	4.52	2.73
M4	27.92	0.1963	416.69	3.19	4.52	2.73
M6	27.93	0.1988	411.69	3.19	4.51	2.73

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
