# Peer review of "Growth of Nanocolumnar TiO2 Bilayer by Direct Current Reactive Magnetron Sputtering in Glancing-Angle Deposition Configuration for High-Quality Electron Transport Layer"

_micromachines, 2023, doi:10.3390/mi14081483_

Round 1

Reviewer 1 Report

This Paper  described "High-quality ETL made by nanocolumns of TiO2 through DC reactive magnetron sputtering in GLAD configuration".

For the acceptance, this paper should be  revised in following points;

1.  The relationship between ETL properties and the results of this paper.

2. In discussion, the structure should be revised to easy understanding; effect of  sputter power and effect of gas ratio, etc.

3. In figure, eg, use Fig. 3(a), (b) and (c). Not above bottom, middle in Fig.    

4. It is better to read and follow author introduction. 

Read carefully again your paper: bi layer => bilayer etc.  

Author Response

All comments have been attended to as far as possible, and the observations and suggestions provided to improve the article were attended to in a timely manner.

Thank you very much!

Reviewer 2 Report

This work offers an accurate report of nanocolumnal growth of TiO2 in the anatase phase using DC sputtering, where such nanocolumns offer the essential characteristics that a high-quality ETL requires, which complements the research gaps of magnetron sputtering technology in the preparation of nanomaterials and is innovative. However, the article still has some issues, and it needs to be reconsidered after major revision:

1.       In line 119 on page 4, “the structure 2 X-ray diffraction analysis performed on samples can be observed from the diffraction patterns”, please indicate the location of “Structure 2” in the corresponding image.

2.       The sample obtained at different powers in Figure 2 has the difference in diffraction peak shift and diffraction intensity, please analyze the reasons in detail.

3.       Figure 3 still has a characteristic peak on 1250 cm-1 for 100 W and 120 W samples, but not for 140 W samples, please indicate what the characteristic peak at this location is related to, and analyze the cause of the difference.

4.       Please provide a detailed description of what is in table 2 in the manuscript and analyze the reasons for the discrepancy in the crystallographic data of M3-6.

5.       On page 5, line 150 mentions “The growth of Tauc's curve becomes linear when the photon energy coincides with the value of the band gap”, please analyze further what does this linear growth trend indicate?

6. Please label the group images in Figure 8 for ease of interpretation in the text and add a scale bar to the top right picture as the left picture is not a profile image at the same magnification. In addition, the inclination angle of the columnar crystals in Figure 8 does not match the inclination angle of the columnar crystals in Figure 70 marked in the figure, obviously the columnar crystals with the horizontal direction greater than <>°, please correct.

Please optimize the illustrations in the article to make the color and layout more beautiful. Please check the manuscript carefully for formatting problems and clerical errors and correct them according to the appropriate standards.

Author Response

All the comments made to the article have been addressed, we appreciate your time to help us improve.

Thak you very much!

Reviewer 3 Report

After going through this manuscript, many statements are unclear. The data presented is not reflected with the description in part 3 and no inline with part 4. There are too many errors can be observed. This manuscript does not representing a good quality work and hence it could not be not recommended.

·         Abstract too brief, less scientific major finding were report.

·         Introduction part, this work is focus on the fabrication of TiO2 film via sputtering, author did not add some review related to deposition parameter or condition that affect the quality of films.

·         Materials and methods part, line 65, “…following proportions: 5/40, 10/40, 15/40 sccm…”, table 1 mention first layer on deposited at 5/40 and 15/40, which one is correct.

·         Materials and methods part, line 67, “…namely glass and silicon (100)…”, what type of glass is used?

·         Materials and methods part, line 67, method mention two types of substrate, but in the results, only show or mention one types?

·         Materials and methods part, line 74, “…power of deposition was 120W and 140W,…”, in table 1, only 120W used.

·         Results part, line 99, “…observe three regions,…”, in Figure 1, the region is hard to see. Need to add a marker, line or drawing to show where the layer is and columnar located. The colour contrast is not nice and clear, cannot see the mention features.

·         Figure 2 & 3, the graph legend used donot know which sample is refer to. Table 1 only mention 120W and 140W.

·         Figure 2, XRD peak of (220) should be representing (211).

·         Figure 3, range selected (x-axis) resulting that the peak of O-Ti-O and N-TiO2 unable to see clearly.

·         Results part, line 113, “…deposit power to 100 W, 120 W and 140 W, …”, this is not similar to table 1.

·         Results part, line 114, “… an angle 2θ = 25.4 as principal maximum…”, angle mention is not similar location as in Figure 2.

·         Results part, line 118, “Regarding the effect of the deposit power…”, this statement unable to understand.

·          Results part, line 124, “…number of 734 cm−1 and a bond of N − TiO2 for a wavenumber of 1106 124 cm−1 [31].”, Paper [31] did not mention peak related to N-TiO2 peak. The value corresponding to Ti-O-Ti bond is 640 cm−1.

·         Figure 4, 5, 6, 7, 9, symbol use in the legend make reviewer have a hard time to co-relate with the correct data.

·         Refer Figure 4 & 5, The XRD peak shifting inconsistence. Compare sample M4 with M3 & M6, peak (004) shifts a lot compared to peak (101) & (200), this might be suggested that XRD spectrum might be belong to others compound.

·         Refer Figure 4 & 5, Compared XRD of sample M3 and M6, the ratio of (004) and (200) peak is far difference, and refer to IR spectra, O-Ti-O speak unable to see, hence, it might be indicating that the XRD spectra obtained for M6 sample might not be representing TiO2 phase.

·         Refer table 2, from Fig.4, M4, peak (004) shifts a lot, but a,b,c still remains close, it is not logic.

·         Results part, line 137, “…while the lattice parameters were calculated using Bragg’s law.”, statement wrong.

·         Results part, line 144, “…which 144 corresponds to the O-Ti-O bond of TiO2 [37].”, reference [37] only mention about Si–O–Ti, there is not discussion and results related to O-Ti-O.

·         Results part, line 147, “…the maximum transmission percentage oscillates between 90% and 95%.”, Maximum take place at around 450nm and 800nm only. From Fig. 6, can not see any value oscillates between 90% and 95%.

·         Results part, line 159, “The upper left image corresponds … the second layer is observed.”,  Fig.8, upper left image surface of nanocolumns cannot see islands profile at the surface (the surface is nice and well flat), why cannot see the island profile?

·         Results part, line 166, “…filaments which correspond to the height of the nanocolumns.”, filaments is in nm scale, the image is in micron scale, so, cannot see clearly.

·         Results part, line 169, “…70 degrees, nanocolumns inclined at values close to this angle are being observed. The measured angles in the inclination of the nanocolumns range from 68 to 80 degrees.”, can not see clearly from given figure.

·         Results part, line 173, “Figure 10 correspond… diameter of 70 nanometers.”, cannot see clearly, image in Fig.10 looks similar, but results in Fig.11 show there is a different.

·         Discussion, line 185 – 193, Statement unable to verify, due to results given in table 1 and Figure 2 is not consistence.

·         Discussion, line 199 – 202, “In this case, we believe…”, O-Ti-O band is higher in M3, why crystalline quality of sample M6 is better?

·         Discussion, line 210, “…, a strong relationship was found between the deposition angle…”, This work, only use one angle (70 degree), how this statement can be conclude?

·         Discussion, line 211-213, Statement contradict and unclear.

·         Discussion, line 215-218, How to verify the homogeneity?

Conclusion part, need to have a more comprehensive conclusion, give significant finding (value, rang or condition) instate of give common statement.

Author Response

All comments and suggestions have been addressed in a timely manner, we appreciate your support to improve the article.

Thank you very much!

Round 2

Reviewer 1 Report

This paper is much improved compared to first version. Though there are some typo like Tio2, TiO2, etc.

The authors have to revise them.

Author Response

Thank you for your valuable contribution and for being an integral part of the publication process. Your expertise and guidance have been invaluable, and we are truly grateful for your support.

Best regards.

Reviewer 2 Report

It's still needed further revison before the manuscript could be published in “Micromachines ” . At present, I am strongly hoping that all the authors need to be still further revised before this manuscript could be completely accepted.

See more in details at the “Comments and Suggestions for Authors

Author Response

(The authors gave the same response as above.)

Reviewer 3 Report

After reading the revised version of the manuscript, suggestion of corrections and comment are as follow: 

1. Refer table 2, suggested that " Planar thin film " and "Nanocolumnar thin film" replace with "Compact TiO2 layer" and "Nanocolummar TiO2 layer" to inline with the title used in section 3.1 & 3.2. Similar goes to Figure 1.

2.Section 3.1,  Line 150, "The diffractograms in figure 2 shows...", at location 53.7, there is not XRD peak, marker or peak indicate Rutile peak (200). Figure 2 show only (220) Rutile peak at position near 53.5 degree as declare by author.

3. Section 3.1, Line 153, "The three thin layers...", suggest change to "Three test sample pf compact TiO2 layer..." 

4. Section 3.2, line 186, "In addition this sample...",  manuscript mention 5 XRD peak, but the corresponding position given only 4 value.

Author Response

(The authors gave the same response as above.)
